# Raised inflammatory markers as a predictor of one-year mortality: a cohort study in primary care in the UK using electronic health record data

Jessica Watson ,[1] Penny Whiting,[2] Chris Salisbury ,[1] Jonathan Banks ,[2] Willie Hamilton [3]

¹Centre for Academic Primary Care, University of Bristol, Bristol, UK
²Bristol Population Health Science Institute, University of Bristol, Bristol, UK
³University of Exeter Medical School, University of Exeter, Exeter, UK

**Correspondence to**
Dr Jessica Watson;
jessica.watson@bristol.ac.uk

## ABSTRACT

**Objectives** Identification of patients at increased mortality risk is important in the context of increasing multimorbidity and an ageing population, to help facilitate the planning and delivery of services. The aim of this study was to examine 1-year all-cause mortality in a cohort of primary care patients in whom inflammatory markers including C reactive protein (CRP), erythrocyte sedimentation rate (ESR) and plasma viscosity (PV), had been tested.

**Design** Observational cohort study using general practitioner Electronic Health Records from the Clinical Practice Research Datalink, with linkage to Office for National Statistics (ONS) Death Registry.

**Setting** UK Primary Care.

**Participants** 159 325 patients with inflammatory marker tests done in 2014 and 39 928 age, sex and practice-matched controls without inflammatory marker testing. ONS Death registry data were available for 109 966 participants.

**Primary and secondary outcome measures** One-year mortality in those with raised inflammatory markers compared with normal inflammatory markers and untested controls. Subanalyses stratified 1-year mortality by age group, gender and cause of death.

**Results** Patients with a raised inflammatory marker (n=47 797) had an overall 1-year all-cause mortality of 6.89%, compared with 1.41% in those with normal inflammatory markers (p<0.001) and 1.62% in untested controls. A raised CRP is associated with the highest mortality rate at 8.76% compared with 4.99% for ESR and 4.66% for PV. One-year mortality is higher in men with a raised inflammatory marker compared with women (9.78% vs 5.29%). The C-statistic of a simple mortality prediction model containing age, sex and CRP test result is 0.89.

**Conclusions** Inflammatory markers are a strong predictor of all-cause mortality in primary care, with a C-statistic comparable to several previously developed frailty indices. Future research should consider the added value of CRP testing, in combination with other risk factors, to improve prediction of mortality in primary care. Evidence- based interventions for frailty are needed alongside predictive tools.

## BACKGROUND

Identification of patients at increased risk of mortality is important in the context of increasing multimorbidity and an ageing

### Strengths and limitations of this study

► The main strength of this study is its size and setting in primary care, making results relevant to clinical practice.
► As test results were transferred electronically to the general practitioners record, there is a very low risk of transcription error or bias.
► Use of ONS death registry data increased the accuracy of recording of mortality, the primary outcome.
► Main limitation is the lack of information on the reasons for testing.
► Previous studies have focused on predictors of mortality in the elderly, yet we were able to study mortality across all age groups.

population, with the aim of helping facilitate the planning and delivery of services. Multiple risk tools have therefore been developed to predict mortality; these can be used to help predict frailty, unplanned hospital admissions and to allow targeted interventions to people at an increased mortality risk. The National Institute for Health and Care Excellence multimorbidity guidelines systematically reviewed 41 of these mortality risk tools; the majority were of low or very low quality and a need for further research in this area was identified.[1] Current risk tools include variables such as disease status, sociodemographic factors and laboratory test results (eg, anaemia, raised platelets), however, none in current use include an inflammatory marker test.

Inflammatory markers such as C reactive protein (CRP), erythrocyte sedimentation rate (ESR) and plasma viscosity (PV) are commonly used in primary care to aid diagnosis and monitoring of infections and inflammatory conditions. Cohort studies in the general population have reported inflammatory markers as predicting future

mortality,[2–4] particularly from cardiovascular disease.[5] CRP has also been shown to have predictive value for mortality in hospital settings.[6–8] The clinical relevance of these findings in primary care settings, and over the shorter term, has not previously been described.

The aim of this study was to examine 1-year all-cause mortality in a cohort of primary care patients in whom inflammatory marker bloods had been tested.

## METHODS

This was a secondary analysis of an observational cohort study of 160 000 patients aged >18 from the Clinical Practice Research Datalink (CPRD) with inflammatory marker blood testing in 2014, and 40 000 age, sex and practice-matched controls without inflammatory marker testing also in 2014. The methods have been described fully in previous paired papers describing the disease outcomes following inflammatory marker testing.[9 10] Patients were excluded from the analysis if the inflammatory marker test result was missing (n=673) or if results were so abnormal as to be considered spurious (n=2). Linkage to ONS death registry data was available for 109 966. The three inflammatory marker tests studied were CRP, ESR and PV. We defined a raised inflammatory marker using the mean upper limit of normal for laboratories within our study. For CRP this was 6.8 mg/L, for simplicity rounded to 7 mg/L; for PV 1.72 mPa.s. For ESR this mean upper limit of normal was rounded and stratified by gender and age.[9] When the same inflammatory marker was repeated on the same day (n=231), we retained the highest value. The index date was defined as the first date of inflammatory marker testing in 2014, with 1-year mortality defined as death within 1 year of this index date. Date of death was defined as the earlier date of recorded death in either CPRD or ONS death registry. Cause of death was available from death certification data in 3141 out of 5512 deaths where ONS linkage was available.

### Analysis

The primary analysis compared 1-year mortality in those with raised versus normal inflammatory markers and compared with untested controls. Subanalyses stratified 1-year mortality by age group and gender and cause of death. For each of the three tests (CRP, ESR and PV), dichotomised test results were cross-classified with the reference standard 1-year mortality, allowing sensitivity and specificity to be calculated. Logistic regression was used to calculate diagnostic ORs, with and without adjustment for age and gender.

Test results were also analysed on a continuous scale, using logistic regression to determine the dose response relationship between inflammatory marker test result and mortality. The area under receiver operator curve, otherwise known as the C-statistic, was calculated using a logistic regression model using test

result on a continuous scale, with log transformation due to the skewed nature of the data, with and without age and gender as covariates. Comparisons of the C-statistic were made using DeLong method.[11] All analyses were done using Stata V.15.[12]

### Patient and public involvement

Patients were not involved in the design, conduct, reporting or dissemination of this research.

## RESULTS

Demographics of the tested cohort and untested controls, compared with the UK population are shown in online supplemental table 1. After exclusions there were 159 325 tested patients of whom 114 198 (71.7%) had a CRP test, 92 325 (58.0%) an ESR test and 15 994 (10%) a PV test; 62 789 (39.4%) had more than one inflammatory marker performed simultaneously, mostly CRP and ESR together (51 546). Overall 47 797 (30%) had one or more raised inflammatory marker on the index date. In total 5512 patients died within 1 year of the index date; 648 deaths in the untested group, 1572 deaths in the normal inflammatory marker group and 3292 deaths in the group with one or more raised inflammatory marker.

### Overall mortality rates

Table 1 shows overall mortality rates subdivided by age, gender and test results. Patients with a raised inflammatory marker (n=47 797) had an overall 1-year mortality of 6.89%, compared with 1.41% in those with normal inflammatory markers (p<0.001). In the untested comparison cohort, 1-year mortality was 1.62%. The association between raised inflammatory markers and 1-year mortality was seen in all age groups apart from the under 30 years old. In older age groups the absolute increase in risk was considerable; a raised inflammatory marker in the over 80s was associated with a 1-year mortality of 21.8%, compared with 8.6% in the over 80s with normal inflammatory markers.

Men with a raised inflammatory marker had a significantly higher 1-year mortality rate than women with a raised inflammatory marker (9.78% vs 5.29%). Patients with a raised CRP had a 1-year mortality of 8.76% compared with 4.99% for those with raised ESR and 4.66% for raised PV.

In the 62 789 patients with more than one inflammatory marker performed simultaneously on the index date, 1-year mortality was higher in the 9029 patients with concordant raised values at 6.9%, compared with the 13 783 with discordant results (one raised, one normal) who had a 1-year mortality of 2.8%. In the 39 977 patients with two simultaneous negative inflammatory markers 1-year mortality was 0.85%.

Table 2 shows the performance characteristics of inflammatory markers, including sensitivity, specificity and C-statistic. CRP had the highest sensitivity of the three tests at 67.8% and the greatest C-statistic at 0.78. OR

**Table 1** One-year mortality (%, 95% CI) subdivided by age, gender and test result

| | Untested (n=39928) | Normal inflammatory markers (n=111528) | Any raised inflammatory marker (n=47797) | Raised CRP (n=29164) | Raised ESR (n=23138) | Raised PV (n=4568) |
|---|---|---|---|---|---|---|
| Overall (n=199253) | 1.62 (1.50 to 1.75) | 1.41 (1.34 to 1.48) | 6.89 (6.66 to 7.11) | 8.76 (8.43 to 9.08) | 4.99 (4.71 to 5.27) | 4.66 (4.05 to 5.27) |
| Age | | | | | | |
| <30 (n=21732) | 0.09 (0.02 to 0.18) | 0.04 (0.01 to 0.08) | 0.08 (0.00 to 0.17) | 0.13 (0.00 to 0.27) | 0.00 (0.00 to 0.00) | 0.0 (0.00 to 0.00) |
| 30–39 (n=22718) | 0.04 (0.00 to 0.10) | 0.08 (0.03 to 0.13) | 0.37 (0.19 to 5.47) | 0.50 (0.23 to 0.77) | 0.28 (0.06 to 0.51) | 0.0 (0.00 to 0.00) |
| 40–49 (n=31588) | 0.19 (0.08 to 0.30) | 0.15 (0.09 to 0.20) | 1.12 (0.86 to 1.39) | 1.54 (1.14 to 1.95) | 0.92 (0.59 to 1.25) | 0.90 (0.01 to 1.69) |
| 50–59 (n=35044) | 0.41 (0.26 to 0.56) | 0.41 (0.32 to 0.49) | 2.37 (2.03 to 2.70) | 2.98 (2.48 to 3.47) | 1.79 (1.38 to 2.21) | 1.62 (0.75 to 2.50) |
| 60–69 (n=35094) | 0.84 (0.62 to 1.05) | 0.91 (0.78 to 1.05) | 4.96 (4.51 to 5.40) | 6.72 (6.05 to 7.39) | 3.88 (3.33 to 4.43) | 3.27 (2.12 to 4.42) |
| 70–79 (30, 251) | 2.27 (1.90 to 2.64) | 2.51 (2.26 to 2.76) | 9.39 (8.77 to 10.0) | 11.38 (10.5 to 12.2) | 7.35 (6.54 to 8.16) | 6.32 (4.75 to 7.88) |
| >80 (22, 826) | 8.88 (8.05 to 9.70) | 8.61 (8.07 to 9.16) | 21.8 (20.9 to 22.7) | 25.9 (24.7 to 27.1) | 16.1 (14.9 to 17.3) | 15.6 (12.9 to 18.3) |
| Gender | | | | | | |
| Male (n=75787) | 1.86 (1.64 to 2.07) | 1.58 (1.46 to 1.70) | 9.78 (9.33 to 10.2) | 11.48 (10.9 to 12.1) | 7.98 (7.35 to 8.61) | 6.63 (5.44 to 7.81) |
| Female (n=123466) | 1.48 (1.33 to 1.63) | 1.30 (1.22 to 1.39) | 5.29 (5.04 to 5.54) | 6.99 (6.61 to 7.36) | 3.65 (3.36 to 3.94) | 3.51 (2.84 to 4.18) |

CRP, C reactive protein; ESR, erythrocyte sedimentation rate; PV, plasma viscosity.

reduced after adjustment for age and gender but were still significant with an adjusted OR for a raised CRP of 4.5 (p<0.001), 2.9 for raised ESR and 2.1 for raised PV.

A logistic regression model containing age (as a continuous variable) and gender had a C-statistic of 0.85, compared with 0.89 for a full model containing age, gender and CRP test result as a continuous variable (p<0.001); 0.88 with age, gender and ESR (p<0.001); and 0.87 with age gender and PV (p<0.001).

### Repeat testing

Figure 1 shows the 1-year mortality in patients according to the subsequent repeat inflammatory marker results, using the most common test performed; CRP. The fact that a CRP test was requested by a general practitioner (GP) was in itself, predictive of increased mortality, with 1-year mortality of 3.3% in the tested vs 1.6% in the untested cohort. This increased to 8.76% 1-year mortality if a single CRP test was raised, 9.13% if a second test was persistently raised and 14.5% if the second test was raised further still. Those with a raised inflammatory marker which was not subsequently rechecked had a 1-year mortality rate of 10.2%, compared with 3.25% if a subsequent CRP normalised.

### Dose–response relationship

A dose–response relationship was found between result of the index CRP test as a continuous variable and 1-year mortality (figure 2). In 2184 people with a CRP ≥100 mg/L overall 1-year mortality was 20.2%. Similar associations, with wider CIs, were found for ESR and PV (not shown).

### Cause of death

Cause of death from ONS death certification was available for 3141 out of 5512 total deaths in the cohort. Table 3 summarises the cause of death among patients with raised inflammatory markers, compared with those with normal inflammatory markers and untested controls. The most common cause of death in the 26 507 patients with raised inflammatory markers was cancer (696 deaths), followed by cardiovascular disease (449 deaths). Odds of mortality in the raised versus normal inflammatory marker groups was highest for cancer (adjusted OR 6.34), followed by infections (adjusted OR 4.11). However, significant increased odds of mortality were seen for all disease categories with the exception of deaths due to falls, musculoskeletal causes and senility. Online supplemental table 2 shows cause of death by age group for patients with raised inflammatory markers; cancer was the most common cause of death in 40–79 years old, cardiovascular disease increased with age and was the most common cause of death in the over 80 age group.

### DISCUSSION

Inflammatory markers are a strong predictor of all-cause mortality in primary care. The association between raised inflammatory markers and all-cause mortality is seen in

**Table 2** Performance characteristics of CRP, ESR and PV for predicting 1-year mortality

| | Sensitivity | Specificity | C-statistic* | Univariable logistic regression | | Adjusted for age and gender | |
|---|---|---|---|---|---|---|---|
| | | | | OR | P value | OR | P value |
| CRP | 67.8% (66.3–69.3) | 75.9% (75.6–76.2) | 0.78 (0.77–0.78) | 6.6 (6.2 to 7.1) | <0.001 | 4.5 (4.2 to 4.8) | <0.001 |
| ESR | 56.6% (54.4–58.7) | 75.7% (75.4–75.9) | 0.66 (0.65–0.67 | 4.1 (3.7 to 4.4) | <0.001 | 2.9 (2.7 to 3.2) | <0.001 |
| PV | 52.0% (47.0–56.9) | 72.1% (71.3–72.8) | 0.62 (0.60–0.64) | 2.8 (2.3 to 3.4) | <0.001 | 2.1 (1.7 to 2.6) | <0.001 |

*C-statistic calculated using log transformed test results as a continuous variable.
CRP, C reactive protein; ESR, erythrocyte sedimentation rate; PV, plasma viscosity.

all age groups except patients aged less than thirty years. Men with raised inflammatory markers have a higher 1-year mortality than women (9.78% vs 5.29%). Of the three tests examined, CRP has the highest predictive accuracy for mortality. The overall C-statistic of a model containing age, sex and CRP test result of 0.89 is comparable to several previously developed frailty indices. Inflammatory markers could potentially be a simple indicator to improve prediction of life expectancy in primary care.

### Strengths and limitations

The major strength of this study is its size and its setting in primary care, making results relevant to clinical practice. As test results were transferred electronically to the GP record, there is a very low risk of transcription error or bias. Use of ONS death registry data increased the accuracy of recording of mortality, the primary outcome. Previous studies have focused on predictors of mortality in the elderly, yet we were able to study mortality across all age groups. The fact that mortality is not raised in patients with a normal inflammatory marker suggests that the test result, rather than the clinician's decision to test, is significant.[13]

The main weakness is the lack of information about the reasons for testing; we cannot determine which tests were done for diagnosis, monitoring or non-specific purposes. However, this is also a strength, as it increases the generalisability of the results, which are not limited to specific subgroups of tested patients.

### Comparison to previous literature

Several previous frailty indices have previously been developed, the most commonly used being the electronic Frailty Index)[14] and Qmortality.[15] The former has a C-statistic of 0.76; the latter a C-statistic of 0.85 for women and 0.84 for men. More recent research by Deelen *et al* has used combinations of biomarkers to predict mortality across all age groups; 226 potential biomarkers were selected, but CRP, ESR and PV were not considered.[16] They generated a model using 14 biomarkers with a C-statistic of 0.837: however, of the biomarkers considered, only albumin is available in primary care, limiting the clinical usefulness of their findings. CRP by comparison is a low cost and widely available test. With a C-statistic of 0.78 for CRP alone, and 0.89 for a model including CRP, age and gender, inflammatory markers could be a simple indicator with a comparable accuracy to currently used mortality prediction tools.

The association between CRP and mortality is in keeping with population-based studies examining all-cause mortality[2 17] and cardiovascular mortality,[5] as well as hospital-based studies of patients with specific diseases including COPD,[18] diabetes,[19] chronic kidney disease,[8 20]

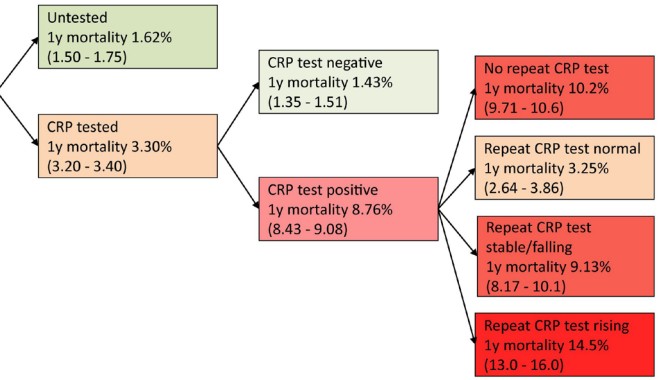

**Figure 1** Flow chart of 1-year mortality (95% CIs) according to CRP test results. The right-hand column shows 1-year mortality according to repeat test result; defined as the first CRP test performed in the 3 months following the index date. CRP, C reactive protein.

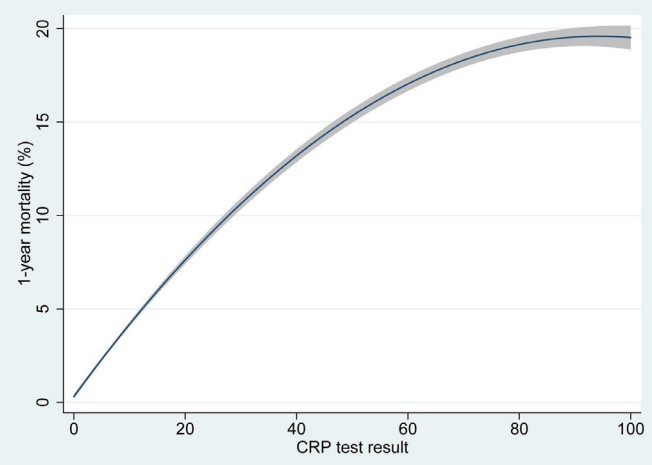

**Figure 2** Polynomial logistic regression of mortality against CRP test result as a continuous variable. CRP, C reactive protein.

**Table 3** Cause of death among patients with ONS death registry linkage (n=109966)

| Cause of death | Untested controls (n=22069) | | Normal inflammatory markers (n=61390) | | Raised inflammatory markers (n=26507) | | Comparison between normal and raised inflammatory markers | |
| | No of deaths | 1-year mortality (%) | No of deaths | 1-year mortality (%) | No of deaths | 1-year mortality (%) | Unadjusted odds ratio (CI) | OR, adjusted for age and gender (CI) |
| --- | --- | --- | --- | --- | --- | --- | --- | --- |
| All-cause mortality | 380 | 1.74 | 889 | 1.45 | 1872 | 7.08 | 5.16 (4.76 to 5.60)* | 3.66 (3.37 to 3.99)* |
| Cancer | 86 | 0.39 | 195 | 0.32 | 696 | 2.63 | 8.46 (7.21 to 9.93) | 6.34 (5.40 to 7.46)* |
| Cardiovascular disease | 115 | 0.52 | 295 | 0.48 | 449 | 1.69 | 3.57 (3.08 to 4.14)* | 2.18 (1.87 to 2.54)* |
| Respiratory | 53 | 0.24 | 141 | 0.23 | 264 | 1 | 4.37 (3.56 to 5.36)* | 2.68 (2.18 to 3.30)* |
| Dementia | 38 | 0.17 | 69 | 0.11 | 119 | 0.45 | 4.01 (2.98 to 5.39)* | 2.21 (1.64 to 2.99)* |
| Gastrointestinal | 16 | 0.07 | 42 | 0.07 | 86 | 0.32 | 4.75 (3.29 to 6.88)* | 3.58 (2.46 to 5.20)* |
| Genitourinary | 8 | 0.04 | 16 | 0.03 | 39 | 0.15 | 5.65 (3.16 to 10.1)* | 3.13 (1.74 to 5.64)* |
| Infection | 5 | 0.02 | 11 | 0.02 | 29 | 0.11 | 6.11 (3.05 to 12.2)* | 4.08 (2.03 to 8.23)* |
| Blood disorder | 2 | 0.01 | 11 | 0.02 | 25 | 0.09 | 5.27 (2.59 to 10.7)* | 3.11 (1.52 to 6.37) |
| Senility | 12 | 0.05 | 8 | 0.01 | 21 | 0.08 | 6.08 (2.69 to 13.7)* | 2.80 (1.23 to 6.38) |
| Musculoskeletal | 3 | 0.01 | 8 | 0.01 | 14 | 0.05 | 4.05 (1.70 to 9.67)* | 2.40 (1.00 to 5.77) |
| Falls | 4 | 0.02 | 10 | 0.02 | 6 | 0.02 | 1.39 (0.51 to 3.82) | 0.80 (0.29 to 2.22) |
| Other | 38 | 0.17 | 83 | 0.14 | 124 | 0.47 | 3.47 (2.62 to 4.59)* | 2.41 (1.82 to 3.20)* |

*P<0.05.

pneumonia[6] and cancer.[21 22] The research reported here demonstrates that this association is also seen in a primary care setting and over the shorter term. The finding that men with raised inflammatory markers are at higher mortality risk than women may reflect gender differences in healthcare-seeking behaviour in primary care; men have lower rates of consultation, so might be 'sicker' on average when presenting for blood tests.

Previous research has shown limited diagnostic utility of inflammatory markers in a primary care setting, where sensitivity is low, false positives are common, and abnormal tests can lead to increased rates of GP consultations, tests and referrals.[9] The fact that inflammatory markers have a higher C-statistic for mortality than for cancer, infections or autoimmune diseases,[10] may reflect the fact that inflammatory markers have both diagnostic and prognostic utility for a broad range of pathologies.

### Clinical implications

GPs should interpret raised inflammatory markers within the wider clinical context; where the cause of inflammation is identifiable and treatable, mortality risks should not cause undue alarm. The findings back up the current clinical practice of repeating an abnormal test; a subsequent normal result is reassuring with mortality risk reducing to near normal. However, clinicians should consider whether older patients with a persistently raised inflammatory markers are reaching the end of life.

There is debate over the utility of predicting mortality,[23] given the lack of evidence-based interventions. We would not recommend that clinicians test inflammatory markers purely for the purpose of mortality prediction, particularly given that false positives have been shown to lead to cascades of follow on tests, appointments and referrals.[9] However, GPs are already required to identify patients who are frail,[24] and inflammatory marker tests are commonly performed for many other reasons. Inflammatory marker test results, when available, may therefore add useful information to improve prediction of mortality and assessment of frailty in primary care.

### Unanswered questions and future research

Future research should consider the added value of CRP testing in combination with other risk factors, to improve prediction of mortality in primary care. Predicting mortality in itself, however, is not enough, as tests will only benefit patients if they influence management. Evidence-based interventions for frailty must therefore be developed alongside predictive tools.

**Contributors** WH, CS, PW, JB and JW all contributed to conception, design and planning of the study. JW analysed the data as part of her Doctoral Research Fellowship, with oversight of the conduct of the study provided by her PhD supervisors WH, CS, PW and JB. JW wrote the first draft of the paper. WH, CS, PW, JB and JW all contributed to subsequent drafts and read and approved the final manuscript.

**Funding** This report is independent research arising from JW's Doctoral Research Fellowship (DRF-2016-09-034) supported by the National Institute for Health Research. JB, PW, CS and JW were supported by the National Institute for Health Research Applied Research Collaboration West (NIHR ARC West). CS is an NIHR Senior Investigator.

**Disclaimer** The views expressed in this publication are those of the authors and not necessarily those of the NHS, the National Institute for Health Research, Health Education England or the Department of Health and Social Care.

**Competing interests** None declared.

**Patient consent for publication** Not required.

**Ethics approval** This study was approved by the Independent Scientific Application Committee for MHRA database research (ISAC) protocol number 17_003. The protocol is available at: https://www.cprd.com/protocol/diagnostic-utility-inflammatory-markers-primary-care-prospective-cohort-study.

**Provenance and peer review** Not commissioned; externally peer reviewed.

**Data availability statement** This study is based on data from the Clinical Practice Research Datalink (CPRD) obtained under licence from the UK Medicines and Healthcare products Regulatory Agency, which does not permit data shaing. The data are provided by patients and collected by the NHS as part of their care and support. The interpretation and conclusions contained in this study are those of the authors alone.

**ORCID iDs**
Jessica Watson http://orcid.org/0000-0002-8177-6438
Chris Salisbury http://orcid.org/0000-0002-4378-3960
Jonathan Banks http://orcid.org/0000-0002-3889-6098
Willie Hamilton http://orcid.org/0000-0003-1611-1373

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
