## [Reviewer comments · BMJ Open]

ARTICLE DETAILS

TITLE (PROVISIONAL)	Raised inflammatory markers as a predictor of one-year mortality: A cohort study in primary care in the United Kingdom using electronic health record data
AUTHORS	Watson, Jessica; Whiting, Penny; Salisbury, Chris; Banks, Jonathan; Hamilton, Willie

VERSION 1 – REVIEW

REVIEWER	Steve Nicholls Monash University, Australia
REVIEW RETURNED	04-Feb-2020

GENERAL COMMENTS	In this manuscript, Watson and colleagues have investigated the association between inflammatory markers and all cause mortality rates in a large general practice database. They report that those individuals with an elevated CRP level have a significantly greater mortality rate than those with normal CRP levels and those in whom inflammatory markers were not measured. The findings are interesting and contribute further to a large body of literature associating inflammatory markers (particularly CRP) and adverse clinical outcomes. The use of a large general practice database with good data linkage are a particular strength. The authors should consider additional points. 1. The novelty is not particularly great.2. The lack of information regarding why the tests were performed and how management was changed is a major limitation.3. Medication use would be helpful4. The authors would be advised to not use abbreviations in the title that are not universally known by all readers.5. How are these findings supposed to be integrated into clinical practice? Where is the data that measuring such parameters alters clinical management and outcome?
---

REVIEWER	Cherian Varghese World Health Organization, Geneva, Switzerland.
REVIEW RETURNED	23-Feb-2020

GENERAL COMMENTS	1. This study adds value in the field of risk markers to predict adverse outcomes.2. Reason for asking for a test can e better explained. Is it an usual practice? What was the guidance provided to GPs to indicate the need for the tests.3. Were all three tests done routinely?4. Did the non-tested people vary in some parameters?5. Can the predictive probability increase if all three test were positive?
---

	6. Are there any differences in prediction of cancer versus CVD deaths in subgroups? Age group? Pre existing disease? 7. Longer term follow up will be helpful. One year mortality indicates the possibility of existing disease conditions.
--	--

REVIEWER	Dr Amy Rogers University of Dundee
REVIEW RETURNED	06-Mar-2020

GENERAL COMMENTS	Dr Amy Rogers University of Dundee 06-Mar-2020 Thank you for inviting me to review this CPRD cohort study evaluating the relationship between inflammatory marker testing, results and one-year mortality in an unselected primary care population. I have only a few suggestions for improvement: Abstract It is not clear from the existing Objectives section of the abstract why demonstrating a link between inflammatory marker testing, results and mortality might be useful. Can this section be extended with a background sentence explaining this? Page 6 lines 3-6 - The relationship between this study and the one cited in reference 9 is not clear without reading reference 9. Please clarify in the text Page 6 line 14 - "normal laboratories within our study." Please clarify what you mean by "normal laboratories". Page 6 line 17 - "6.8mg/l, for simplicity rounded". Why was it considered necessary to round the CRP cut-off but not PV or ESR? Please justify this decision. Page 9 line 56 - It may be clearer to refer to first, initial or index CRP result here. Page 11 lines 50-57 - "The association between raised inflammatory markers and all-cause mortality is seen in all age groups except the under thirties, with higher one-year mortality in men with raised inflammatory markers than women (9.78% vs 5.29%)". As I read it, this sentence implies that the sex difference in mortality was only present in the under 30's. If this is not the case, I would suggest splitting into two separate sentences for clarity. Minor grammatical/typographical issues: Page 5 line 25 - is the colon after the brackets intentional? (Colons are more usually used to separate two independent clauses where the second explains or illustrates the first.) Page 9 line 31 - "persistently raised and 14.5% the second test was raised" Is there an "if" missing between "14.5%" and "the"? Page 10 line 31 - "odds of mortality was seen" should be "were seen"?
---

	Page 13 line 28 - "...sicker' on average when receiving blood tests". Should this be "when presenting for blood tests"? Data sharing statement Missing "r" in sharing on line 35.
--	---

VERSION 1 – AUTHOR RESPONSE

Reviewer: 1

In this manuscript, Watson and colleagues have investigated the association between inflammatory markers and all cause mortality rates in a large general practice database. They report that those individuals with an elevated CRP level have a significantly greater mortality rate than those with normal CRP levels and those in whom inflammatory markers were not measured. The findings are interesting and contribute further to a large body of literature associating inflammatory markers (particularly CRP) and adverse clinical outcomes. The use of a large general practice database with good data linkage are a particular strength.

Thank you

The authors should consider additional points.

1. The novelty is not particularly great.

Although we are aware of the significant amount of previous research into CRP, including cohort studies looking at cardiovascular disease and mortality and hospital based studies of CRP and mortality, the clinical relevance of a raised inflammatory marker in primary care settings, and over the shorter term, has not previously been described as far as we can tell.

2. The lack of information regarding why the tests were performed and how management was changed is a major limitation.

We acknowledge this limitation in the discussion – unfortunately it is not possible to overcome this as the reasons for testing are not necessarily recorded in routine medical records and are therefore unavailable to researchers in the Clinical Practice Research Datalink. We would counter that this does increase the generalisability of the results, which are not limited to specific subgroups of tested patients. It also reflects clinical practice, as GPs often cross-cover for colleagues, trainees and locums, and therefore the reason for testing is not always apparent when test results are reviewed.

3. Medication use would be helpful

Although looking at medications could be useful to explore additional research questions, this was not relevant to the research questions addressed in this paper.

4. The authors would be advised to not use abbreviations in the title that are not universally known by all readers.

We have changed the title and removed abbreviations:

Raised inflammatory markers as a predictor of one-year mortality: A cohort study using primary care electronic health record data

5. How are these findings supposed to be integrated into clinical practice? Where is the data that measuring such parameters alters clinical management and outcome?

Whilst the findings are interesting, we fully agree with the reviewers comments and would not recommend GPs measure inflammatory markers for the purpose of predicting mortality, as there is no evidence this would alter clinical management and outcomes. However, GPs are already being asked

to measure and predict frailty (<https://www.england.nhs.uk/ourwork/clinical-policy/older-people/frailty/frailty-risk-identification/>) using the electronic frailty index which has a similar c-statistic to a single CRP test. The clinical relevance is therefore to improve prediction modelling which is already happening in clinical practice. We have added an extra sentence to clarify this.

We would not recommend that clinicians test inflammatory markers purely for the purpose of mortality prediction, particularly given that false positives have been shown to lead to cascades of follow on tests, appointments and referrals.¹ However, general practitioners are already required to identify patients who are frail, and inflammatory marker tests are commonly performed for many other reasons. Inflammatory marker test results, when available, may therefore add useful information to improve prediction of mortality and assessment of frailty in primary care.

Reviewer: 2

Please leave your comments for the authors below

1. This study adds value in the field of risk markers to predict adverse outcomes.

Thank you

2. Reason for asking for a test can be better explained. Is it usual practice? What was the guidance provided to GPs to indicate the need for the tests.

This was an observational study, using routinely collected data from primary care electronic health records. There were no instructions or guidance to GPs and we simply observed routine clinical practice. We have added observational cohort study to the first line of the methods and abstract to clarify this.

3. Were all three tests done routinely?

We have added an additional sentence to clarify how many patients had more than one test simultaneously.

After exclusions there were 159,325 tested patients of whom 114,198 (71.7%) had a CRP test, 92,325 (58.0%) an ESR test, and 15,994 (10%) a PV test; 62,789 (39.4%) had more than one inflammatory marker performed simultaneously, mostly CRP and ESR together (51,546).

4. Did the non-tested people vary in some parameters?

Non-tested people were matched by age, sex and practice to the tested population. We have added summary demographics as an additional supplementary table 1 and an additional sentence to the results:

Demographics of the tested cohort and untested controls, compared to the UK population are shown in supplementary table 1.

5. Can the predictive probability increase if all three test were positive?

Only 403 patients had all three tests performed simultaneously, so we could not examine this question exactly, however we have added a section detailing the predictive probability for patients with more than one test performed simultaneously:

In the 62,789 patients with more than one inflammatory marker performed simultaneously on the index date, one-year mortality was higher in the 9,029 patients with concordant raised values at 6.9%, compared to the 13,783 with discordant results (one raised, one normal) who had a one-year mortality of 2.8%. In the 39,977 patients with two simultaneous negative inflammatory markers one-year mortality was 0.85%.

6. Are there any differences in prediction of cancer versus CVD deaths in subgroups? Age group? Pre existing disease?

Thank you for this suggestion. We have added supplementary table 2 which shows cause of death in patients with a raised inflammatory marker subdivided by age group and added an additional sentence:

Supplementary table 2 shows cause of death by age group for patients with raised inflammatory markers; cancer was the commonest cause of death in 40-79 year olds, cardiovascular disease increased with age and was the commonest cause of death in the over 80 age group.

7. Longer term follow up will be helpful. One-year mortality indicates the possibility of existing disease conditions.

Unfortunately, due to the nature of the dataset we do not have long term follow-up data.

Reviewer: 3

Please leave your comments for the authors below

Thank you for inviting me to review this CPRD cohort study evaluating the relationship between inflammatory marker testing, results and one-year mortality in an unselected primary care population.

I have only a few suggestions for improvement:

Abstract

It is not clear from the existing Objectives section of the abstract why demonstrating a link between inflammatory marker testing, results and mortality might be useful. Can this section be extended with a background sentence explaining this?

Add additional sentence to clarify this:

Identification of patients at increased mortality risk is important in the context of increasing multimorbidity and an ageing population, to help facilitate the planning and delivery of services.

Page 6 lines 3-6 - The relationship between this study and the one cited in reference 9 is not clear without reading reference 9. Please clarify in the text

Thank you, we have clarified:

This was a secondary analysis of a cohort study of 160,000 patients aged >18 from the Clinical Practice Research Datalink (CPRD) with inflammatory marker blood testing in 2014, and 40,000 age, sex and practice matched controls without inflammatory marker testing also in 2014. The methods have been described fully in previous paired papers describing disease outcomes following inflammatory marker testing.^{1,10}

Page 6 line 14 - "normal laboratories within our study." Please clarify what you mean by "normal laboratories".

Apologies this referred to the mean upper limit of normal, the word 'for' was missing from the sentence:

We defined a raised inflammatory marker using the mean upper limit of normal for laboratories within our study.

Page 6 line 17 - "6.8mg/l, for simplicity rounded". Why was it considered necessary to round the CRP cut-off but not PV or ESR? Please justify this decision.

For plasma viscosity the upper limit of normal was 1.72 in 99.85% of cases so there was no justification to round this figure to 1.70. For CRP and ESR the upper limit of normal varied between laboratories so we used the mean upper limit of normal rounded to the nearest whole number as this was simpler and therefore more clinically meaningful.

We have clarified that ESR cut-off was indeed rounded:

For ESR this mean upper limit of normal was rounded and stratified by gender and age.¹

Page 9 line 56 - It may be clearer to refer to first, initial or index CRP result here.

Changed as suggested: A dose response relationship was found between result of the index CRP test

Page 11 lines 50-57 - “The association between raised inflammatory markers and all-cause mortality is seen in all age groups except the under thirties, with higher one-year mortality in men with raised inflammatory markers than women (9.78% vs 5.29%)”. As I read it, this sentence implies that the sex difference in mortality was only present in the under 30’s. If this is not the case, I would suggest splitting into two separate sentences for clarity.

We have clarified by splitting this sentence into two:

The association between raised inflammatory markers and all-cause mortality is seen in all age groups except patients aged less than thirty years. Men with raised inflammatory markers have a higher one-year mortality than women (9.78% vs 5.29%).

Minor grammatical/typographical issues:

Page 5 line 25 - is the colon after the brackets intentional? (Colons are more usually used to separate two independent clauses where the second explains or illustrates the first.)

Thank you, colon removed as suggested.

Page 9 line 31 - “persistently raised and 14.5% the second test was raised” Is there an “if” missing between “14.5%” and “the”?

Yes – quite right – we have added the missing ‘if’.

Page 10 line 31 - “odds of mortality was seen” should be “were seen”?

Thank you, changed as suggested

Page 13 line 28 - “...sicker’ on average when receiving blood tests”. Should this be “when presenting for blood tests”?

We have made this change.

Data sharing statement

Missing “r” in sharing on line 35.

Corrected – thank you.

VERSION 2 – REVIEW

REVIEWER	Varghese, Cherian World Health Organization, Geneva, Switzerland.
REVIEW RETURNED	01-Apr-2020
GENERAL COMMENTS	Authors have provided response to all comments.

VERSION 2 – AUTHOR RESPONSE

Thank you, please find attached a revised document. We have changed the strengths and limitation section and checked the formatting is in accordance with the instructions for authors as requested.

Please let us know if any further adjustments are required.